# Efficacy of SARS-CoV-2 vaccination in patients with monoclonal gammopathies: A cross sectional study

Eugenia Abella[1,2] , Macedonia Trigueros[3], Edwards Pradenas[3] , Francisco Muñoz-Lopez[3], Francesc Garcia-Pallarols[1], Randa Ben Azaiz Ben Lahsen[1], Benjamin Trinité[3], Victor Urrea[3] , Silvia Marfil[3], Carla Rovirosa[3], Teresa Puig[3], Eulàlia Grau[3], Anna Chamorro[4], Ruth Toledo[4], Marta Font[4], Dolors Palacín[5], Francesc Lopez-Segui[4] , Jorge Carrillo[3,6] , Nuria Prat[5] , Lourdes Mateu[4,7,8,9], Bonaventura Clotet[3,4,6,8], Julià Blanco[3,6,8,10], Marta Massanella[3,8,6] , VAC-COV-GM-HMAR, KING Cohort Extension and CoronAVI@S studies

SARS-CoV-2 vaccination is the most effective strategy to protect individuals with haematologic malignancies against severe COVID-19, while eliciting limited vaccine responses. We characterized the humoral responses following 3 mo after mRNA-based vaccines in individuals at different plasma-cell disease stages: monoclonal gammopathy of undetermined significance (MGUS), smoldering multiple myeloma (SMM), and multiple myeloma on first-line therapy (MM), compared with a healthy population. Plasma samples from uninfected MM patients showed lower SARS-CoV-2–specific antibody levels and neutralization capacity compared with MGUS, SMM, and healthy individuals. Importantly, COVID-19 recovered MM individuals presented significantly higher plasma neutralization capacity compared with their uninfected counterparts, highlighting that hybrid immunity elicit stronger immunity even in this immunocompromised population. No differences in the vaccine-induced humoral responses were observed between uninfected MGUS, SMM and healthy individuals. In conclusion, MGUS and SMM patients could be SARS-CoV-2 vaccinated following the vaccine recommendations for the general population, whereas a tailored monitoring of the vaccine-induced immune responses should be considered in uninfected MM patients.

## Introduction

Cancer patients present substantial immune impairment induced by the tumour itself or by their treatment, and therefore they are at increased risk for infections and infection-related mortality. In fact,

SARS-CoV-2 infection in non-vaccinated cancer patients with immunosuppression has been associated with significantly higher morbidity and mortality rates (Zhang et al, 2020), especially those patients with haematological neoplasms (Vijenthira et al, 2020). Patients with plasma-cell dyscrasias, such as multiple myeloma, are associated with immunosuppression and at increased risk of infections due to various circumstances: active disease status, decrease in non-clonal immunoglobulins (immunoparesis) and impairment of cellular immunity, but also can be affected by comorbidities and older age (Dumontet et al, 2018). In addition, new treatment approaches with proteasome inhibitors, immunomodulators, monoclonal antibodies and CAR T cells may also exacerbate this immune dysfunction (Chari et al, 2020), putting these individuals at higher risk of any infection. Indeed, at least 80% of multiple myeloma patients required hospital admission after SARS-CoV-2 infection (Engelhardt et al, 2020), and more than 30% died because of COVID-19 (Chari et al, 2020). SARS-CoV-2 vaccination is the most effective strategy to protect this vulnerable population against severe COVID-19; however, previous studies showed reduced humoral responses against several vaccines (i.e., pneumococci, staphylococcal α toxin, tetanus, among others) (Karlsson et al, 2011).

Given their greater susceptibility to severe COVID-19 and lower vaccine-induced immune responses, patients with monoclonal gammopathies are a high-priority group for vaccination to mitigate COVID-19 related morbidity and mortality (Ribas et al, 2021) and further characterization of their immunity generated are required. Current guidelines recommend SARS-CoV-2 vaccination of all patients with monoclonal gammopathy of undetermined significance (MGUS), smoldering multiple myeloma (SMM), and multiple myeloma on therapy (MM) (Ludwig et al, 2021). It is important to determine the protection achieved with the standard vaccination to

[1]Department of Hematology, Hospital del Mar-IMIM, Barcelona, Spain   [2]Pompeu Fabra University, Barcelona, Spain   [3]IrsiCaixa-AIDS Research Institute, Hospital Universitari Germans Trias i Pujol, Campus Can Ruti, Badalona, Spain   [4]Fundació Lluita Contra les Infeccions, Hospital Universitari Germans Trias i Pujol, Campus Can Ruti, Badalona, Spain   [5]Direcció d'Atenció Primària–Metropolitana Nord, Sabadell, Spain   [6]Centro de Investigación Biomédica en Red de Enfermedades Infecciosas, CIBERINFEC, Madrid, Spain   [7]Infectious Diseases Department, Hospital Universitari Germans Trias i Pujol, Campus Can Ruti, Badalona, Spain   [8]University of Vic–Central University of Catalonia (UVic-UCC), 08500, Vic, Spain   [9]Centro de Investigación Biomédica en Red de Enfermedades Respiratorias, CIBERES, Madrid, Spain   [10]Germans Trias i Pujol Research Institute (IGTP), Campus Can Ruti, Badalona Barcelona, Spain

Correspondence: mabella@psmar.cat; mmassanella@irsicaixa.es

adapt the vaccination calendar to their immune needs and prioritize booster doses administration. Published data about SARS-CoV-2 vaccination efficacy refer mainly to patients with MM and, to a lesser extent, with SMM (Chung et al, 2021; Pimpinelli et al, 2021; Van Oekelen et al, 2021; Terpos et al, 2021a). Data in patients with MGUS are scarce and requires further investigation.

The aim of this study was to evaluate the COVID-19 vaccine humoral responses differ in patients with monoclonal gammopathies, including MGUS, SMM, and MM, compared with a healthy control population, to adjust the vaccination calendar to the subtype of disease.

# Results

## Characteristics of the individuals with monoclonal gammopathies

We recruited 59 patients with monoclonal gammopathies, who were subgrouped according to their disease stage (MGUS, SMM, and MM). We took advantage of the national vaccination plan in cancer patients to evaluate the immune responses generated by the BNT16b2 or mRNA-1273 COVID-19 vaccines. Six patients were excluded because they had received AstraZeneca or Janssen vaccination (Fig S1). All samples were collected after a median of 3.9 IQR [2.2–5] months from complete schedule of vaccine administration (two doses). Additional characteristics of patients are presented in Table 1.

## Impact of hybrid immunity in immune responses to vaccine

Despite none of the participants included in this study had a positive SARS-CoV-2 PCR, we identified 2/17 (12%) and 4/14 (29%) patients from MGUS and MM groups, respectively, that showed anti-NP antibodies, suggesting that these individuals were previously infected by SARS-CoV-2. Because the combination of natural infection and vaccine-generated immunity elicit higher specific immune responses (Reynolds et al, 2021; Stamatatos et al, 2021), uninfected and anti-NP+ subjects suffering from monoclonal gammopathies were analysed separately and compared with a healthy control group (CG), matched by age, sex, and time after complete vaccination schedule. Consistent with a previous infection of these individuals, we found a statistically significant increase of circulating SARS-CoV-2–specific IgG and IgA antibodies against S2+RBD in anti-NP+ MM patients compared with the anti-NP– counterparts ($P$ = 0.003 and $P$ = 0.04, respectively, Fig 1A and B). However, anti-NP+ MM group showed significantly lower SARS-COV-2–specific IgG and IgA antibodies compared with the control-infected group ($P$ = 0.02 and $P$ = 0.04, respectively). Low levels of specific IgM were detected in all groups, except for previously infected control group (Fig 1C). Finally, plasma from anti–NP+ MM patients showed statistically increased levels of neutralization compared with their uninfected counterparts (anti-NP–) and uninfected controls ($P$ = 0.01 and $P$ = 0.04, respectively Fig 1D), whereas similar levels of neutralization were observed between anti-NP+ MM and infected control group ($P$ = 0.4). Because of the low number of anti-NP+ MGUS participants, we could not draw any conclusion.

## Humoral responses after 3 mo from vaccination of uninfected individuals

Because of these differences in immune response to vaccine, we excluded those putative COVID-19 recovered individuals from our analysis to specifically determine the vaccine-generated immunity in patients with monoclonal gammopathies. An uninfected control group (CG) was included in our analysis as a reference group matched by age, sex, and time after complete vaccination schedule (Table 1).

All individuals analysed seroconverted after COVID-19 vaccination; however, there were statistically significant differences among groups in the levels of specific anti–SARS-CoV-2 IgG antibodies (Kruskal–Wallis, $P$ = 0.002, Fig 2A). Patients with MM showed significantly lower levels of SARS-CoV-2–specific IgG antibodies compared with all groups ($P$ < 0.006 in all cases, Fig 2A), whereas no differences were observed between MGUS and SMM with control group. Lower levels of specific SARS-CoV-2 IgA antibodies were detected in all groups, and 33% (5/15), 23% (5/22), and 80% (8/10) of MGUS, SMM, and MM did not develop any specific IgA antibodies, respectively (Fisher's exact test $P$ = 0.008, Fig 2B). Again, MM showed significantly lower levels of SARS-CoV-2–specific IgA antibodies compared with CG and MGUS ($P$ = 0.002 and $P$ = 0.04, respectively) and a tendency compared with SMM ($P$ = 0.08). SARS-CoV-2–specific IgM antibodies were almost undetectable in all samples, including healthy CG (Fig 2C). Similarly to the serology results, patients with MM showed a tendency for lower neutralization capacity compared with CG ($P$ = 0.08, 321 [196–449] and 770 [422–1,554] neutralization titer, respectively), whereas no differences in neutralization levels between MGUS or SMM with CG were observed (Fig 2D). Using the lowest 25th percentile on neutralization titer from CG as cut-off (neutralization titer of 420), we found that 7/10 (70%) of MM patients did not reach this neutralization threshold in comparison of the 8/36 (22%) of the CG individuals (Fisher's exact test, $P$ = 0.003). In contrast, MGUS or SMM groups showed comparable percentages of patients below the established cut-off than control group (4/15 [27%] and 4/22 [18%], respectively, $P$ = 0.48 and $P$ = 1.0 compared with CG). We then estimated the proportion of effective neutralizing antibodies among the total SARS-CoV-2–specific IgG, calculated as the ratio of plasma neutralization titer to total SARS-CoV-2 IgG antibodies, as previously described (Trigueros et al, 2022; Pradenas et al, 2022b). We observed an increase of this ratio in patients with MM compared with other groups (Fig 2E and $P$ = 0.02), suggesting that despite the low levels of specific SARS-CoV-2 antibodies in this population, these antibodies seemed to be functional.

Because treatments received by MM group could cause specific immune dysfunction impairing immune response to COVID-19 vaccines, we first compared the levels of specific SARS-CoV-2 IgG antibodies and neutralization capacity in MM patients receiving or not daratumumab (anti-CD38 therapy). No differences were found between groups (Fig S2A and B). Similar results were found when we compared MM individuals who underwent autologous hematopoietic cell transplantation (auto-HSCT) versus no auto-HSCT (Fig S2C and D). Despite the limited number of patients, these results suggest that these treatments might not the main cause of the immune dysfunction observed in these individuals.

**Table 1.  Patient and disease characteristics.**

| | MGUS Uninfected N = 15 | MGUS Infected N = 2 | P-value between infected and uninfected MGUS[a] | SMM Uninfected N = 22 | MM Uninfected N = 10 | MM Infected N = 4 | P-value between infected and uninfected MM[a] | CG Uninfected N = 36 | CG Infected N = 58 | P-value between infected and uninfected CG[a] | P-value between all uninfected groups | P-value between all infected groups |
|---|---|---|---|---|---|---|---|---|---|---|---|---|
| Female, n (%) | 9 (60) | 2 (100) | 0.51 | 15 (68) | 5 (50) | 4 (100) | 0.22 | 21 (58) | 34 (59) | 1.00 | 0.78 | 0.15 |
| Age, years, median [IQR] | 67 [51–83] | 78 [77–78] | 0.51 | 73 [63–80] | 74 [68–79] | 53 [45–83] | 0.23 | 62 [48–77] | 75 [63–82] | **0.004** | 0.61 | 0.32 |
| Gammopathy type, n (%) | | | | | | | | | | | | |
| IgG (Kappa or Lambda) | 8 (53) | 1 (50) | | 14 (64) | 6 (60) | 3 (75) | | NA | NA | | | |
| IgA (Kappa or Lambda) | 3 (20) | 1 (50) | | 5 (23) | 2 (20) | 0 (0) | | NA | NA | | | |
| IgM (Kappa or Lambda) | 4 (27) | 0 (0) | | 2 (9) | 0 (0) | 0 (0) | | NA | NA | | | |
| Bence Jones Kappa | 0 (0) | 0 (0) | | 1 (5) | 1 (10) | 1 (25) | | NA | NA | | | |
| Bence Jones Lambda | 0 (0) | 0 (0) | | 0 (0) | 1 (10) | 0 (0) | | NA | NA | | | |
| Immunoparesis, n (%) | 6 (35) | 1 (50) | 1.0 | 17 (77) | 9 (90) | 3 (75) | 0.10 | NA | NA | | | |
| Total lymphocytes ($\times10^3$), cells/µl, median [IQR] | 2.5 [2.0–3.0] | 1.8 [1.3–2.4] | 0.37 | 1.8 [1.6–2.4] | 1.5 [0.8–2.3] | 1.8 [1.2–2.3] | 0.84 | NA | NA | | | |
| Lymphopenia ($<10^3$ cells/µl), n (%) | 0 (0) | 0 (0) | 1 | 1 (5) | 3 (30) | 0 (0) | 0.5 | NA | NA | | | |
| Treatment | | | | | | | | | | | | |
| No treatment, n (%) | 17 (100) | 2 (100) | | 22 (100) | 0 (0) | 0 (0) | | NA | NA | | | |
| VRD | 0 (0) | 0 (0) | | 0 (0) | 2 (20) | 3 (75) | | | | | | |
| VRD+autoTPH | 0 (0) | 0 (0) | | 0 (0) | 1 (10) | 0 (0) | | | | | | |
| Daratumumab in combination | 0 (0) | 0 (0) | | 0 (0) | 4 (40) | 0 (0) | | | | | | |
| KRD | 0 (0) | 0 (0) | | 0 (0) | 1 (10) | 0 (0) | | | | | | |
| KRD+autoTPH | 0 (0) | 0 (0) | | 0 (0) | 1 (10) | 0 (0) | | | | | | |
| VTD+autoTPH | 0 (0) | 0 (0) | | 0 (0) | 1 (10) | 0 (0) | | | | | | |
| Prednisona | | | | 0 (0) | 0 (0) | 1 (25) | | | | | | |

**Table 1.    Continued**

| | MGUS Uninfected N = 15 | MGUS Infected N = 2 | P-value between infected and uninfected MGUS[a] | SMM Uninfected N = 22 | MM Uninfected N = 10 | MM Infected N = 4 | P-value between infected and uninfected MM[a] | CG Uninfected N = 36 | CG Infected N = 58 | P-value between infected and uninfected CG[a] | P-value between all uninfected groups | P-value between all infected groups |
|---|---|---|---|---|---|---|---|---|---|---|---|---|
| Detectable anti-NP SARS-CoV-2 IgG, n (%) | 0 (0) | 2 (100) | | 0 (0) | 0 (0) | 4 (100) | | 0 (0) | ND | | | |
| Vaccine, n (%) | | | | | | | *0.58* | | | | **0.0004** | *0.19* |
| Pfizer | 13 (87) | 2 (100) | *1.0* | 9 (41) | 3 (30) | 2 (50) | | 29 (81) | 48 (83) | | | |
| Moderna | 2 (13) | 0 (0) | | 13 (59) | 7 (70) | 2 (50) | | 7 (19) | 10 (17) | | | |
| Time from complete vaccine dose to extraction, days | | | | | | | | | | | | |
| Median [IQR] | 97 [42–144] | 65 [64–65] | *0.36* | 123 [67–158] | 80 [62–153] | 111 [77–177] | *0.81* | 101 [89–112] | 92 [78–107] | ***0.007*** | *0.44* | *0.08* |
| Range | 23–166 | 64–65 | | 28–167 | 33–156 | 77–188 | | 33–184 | 29–141 | | | |
| Comorbidities, n (%) | | | | | | | | | | | | |
| HTA | 6 (40) | 2 (100) | | 8 (36) | 7 (70) | 2 (50) | | 10 (28) | 27 (47) | | *0.11* | *0.45* |
| Autoimmune disease | 0 (0) | 0 (0) | | 0 (0) | 0 (0) | 0 (0) | | 0 (0) | 12 (21) | | *1.0* | *0.72* |
| Allergies | 1 (7) | 0 (0) | | 4 (18) | 2 (14) | 0 (0) | | 4 (11) | 5 (9) | | *1.0* | *1.0* |
| Diabetes | 2 (13) | 0 (0) | | 4 (18) | 2 (14) | 0 (0) | | 7 (19) | 3 (5) | | *1.0* | *1.0* |

[a]P-values obtained from Fisher's exact test for categorical variables and Mann–Whitney for continuous variables.

Significant P-values are indicated in bold and all P-values are in italic.

HSCT, hematopoietic stem cell transplantation; VRD, bortezomib, lenalidomide, dexamethasone; KRD, carfilzomib, lenalidomide, dexamethasone; VTD, bortezomib, thalidomide, dexamethasone.

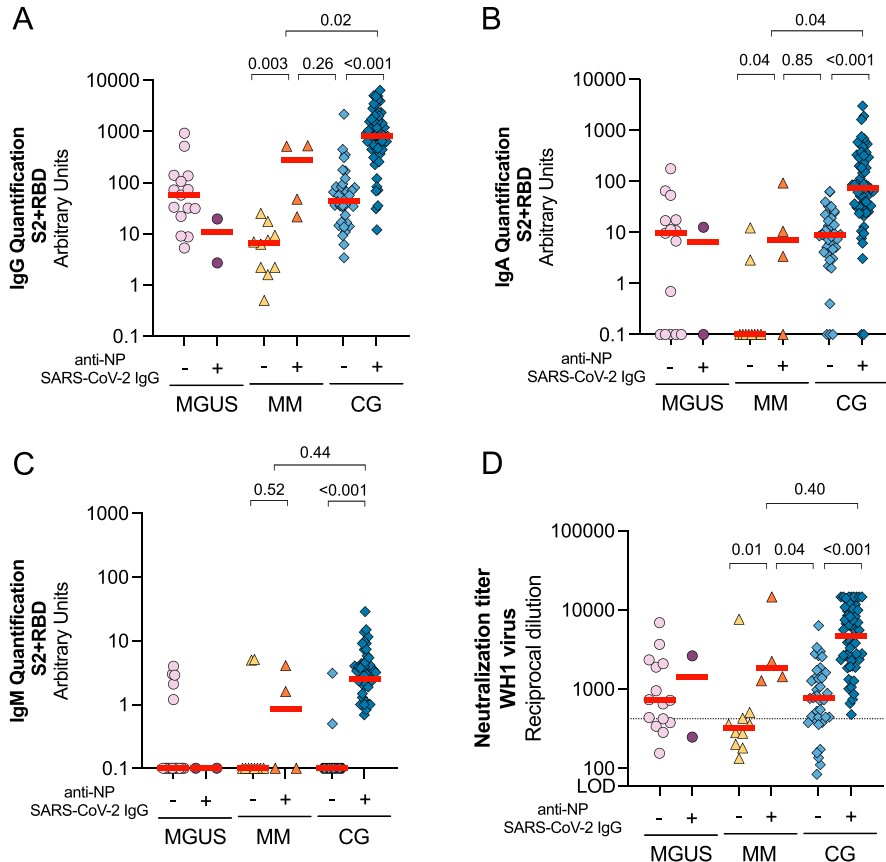

**Figure 1. Comparison of humoral response between uninfected and infected individuals suffering from monoclonal gammopathies compared with a control group (CG).**
**(A, B, C)** Levels of specific SARS-CoV-2 immunoglobulins IgG (Panel A), IgA (Panel B), and IgM (Panel C) against S2+RBD proteins quantified in plasma from uninfected and infected participants by ELISA after 3 mo from SARS-CoV-2 vaccination. Panel **(D)** Neutralizing activity against WH1 virus after 3 mo of vaccine administration in infected and uninfected participants. Dotted line indicates the 25th percentile of neutralization titer from CG. In all panels, median values are indicated and *P*-values were obtained from Mann–Whitney test for comparison between groups. Each symbol represents a participant, and are color-coding according to their disease group (MGUS, purple square; SMM, upper triangle turquoise; MM, lower triangle orange; and CG, blue diamond). Source data are available for this figure.

## Predictive factors for specific SARS-CoV-2 humoral responses

There was a positive correlation between SARS-CoV-2–specific IgG and IgA antibody response and the neutralization titer (r = 0.60 *P* < 0.0001, and r = 0.54 *P* < 0.0001, respectively, Fig 3A–C). On the other hand, we found a negative correlation between the neutralization capacity and the days post-complete vaccine scheduled (r = −0.48, *P* < 0.0001) and age (r = −0.27, *P* < 0.0001). In addition, age was also negatively correlated to the levels of circulating SARS-CoV-2–specific IgG and IgA antibodies (r = −0.37, *P* < 0.0001 and r = −0.34, *P* = 0.002, respectively). Because of the potential impact of different parameters in the humoral response described in the literature and found in our cohort, we performed linear regression models for the levels of specific SARS-CoV-2 IgG and IgA antibodies and neutralization capacity for the variables: group (MGUS, SMM, MM, and CG [reference]), age, sex, immunoparesis (yes/no), vaccine (Pfizer/Moderna), days post complete vaccine schedule, and subtype of gammopathy (IgG, IgA, IgM, and light chain). In the univariate analysis, we found that MM group, age and immunoparesis were significantly associated with lower levels of circulating SARS-CoV-2–specific IgG antibodies (Table 2). In the multivariate analysis, only MM group and age were negative predictors of lower levels of SARS-CoV-2 IgG antibodies. Similarly, MM group, age, IgG gammopathy and immunoparesis were significant predictive factors for lower levels of circulating SARS-CoV-2–specific IgA antibodies in the univariate analysis, and only age remained significant in the multivariate analysis (Table 2). Finally, MM group, age, immunoparesis, and days post-complete schedule (two doses) were significant predictive factors for lower neutralization capacity, whereas Moderna vaccine had a tendency to higher neutralization levels (Table 2). The multivariate analysis revealed that all these parameters remained significant predictors, except for immunoparesis.

Last, we also evaluated the impact of lymphopenia in the humoral response in patients with monoclonal gammopathies. While the total counts of lymphocytes were positively associated with increased levels of specific SARS-CoV-2 IgG antibodies in the univariate linear regression (estimate 0.321, *P* = 0.02), the neutralization capacity was not significant (estimate 0.09, *P* = 0.23).

# Discussion

In this study, we evaluated the SARS-CoV-2 vaccine–induced humoral response in different stages of plasma-cell diseases, revealing that uninfected MM have a lower humoral response to vaccination after 3 mo post-vaccine (two doses) when compared with MGUS and SMM, and a group of healthy controls matched by age and sex. These results are in line with previous published data, which evaluated humoral responses at shorter time points post-SARS-CoV-2 vaccination (Bird et al, 2021; Bitoun et al, 2021; Chung et al, 2021; Pimpinelli et al, 2021; Van Oekelen et al, 2021;

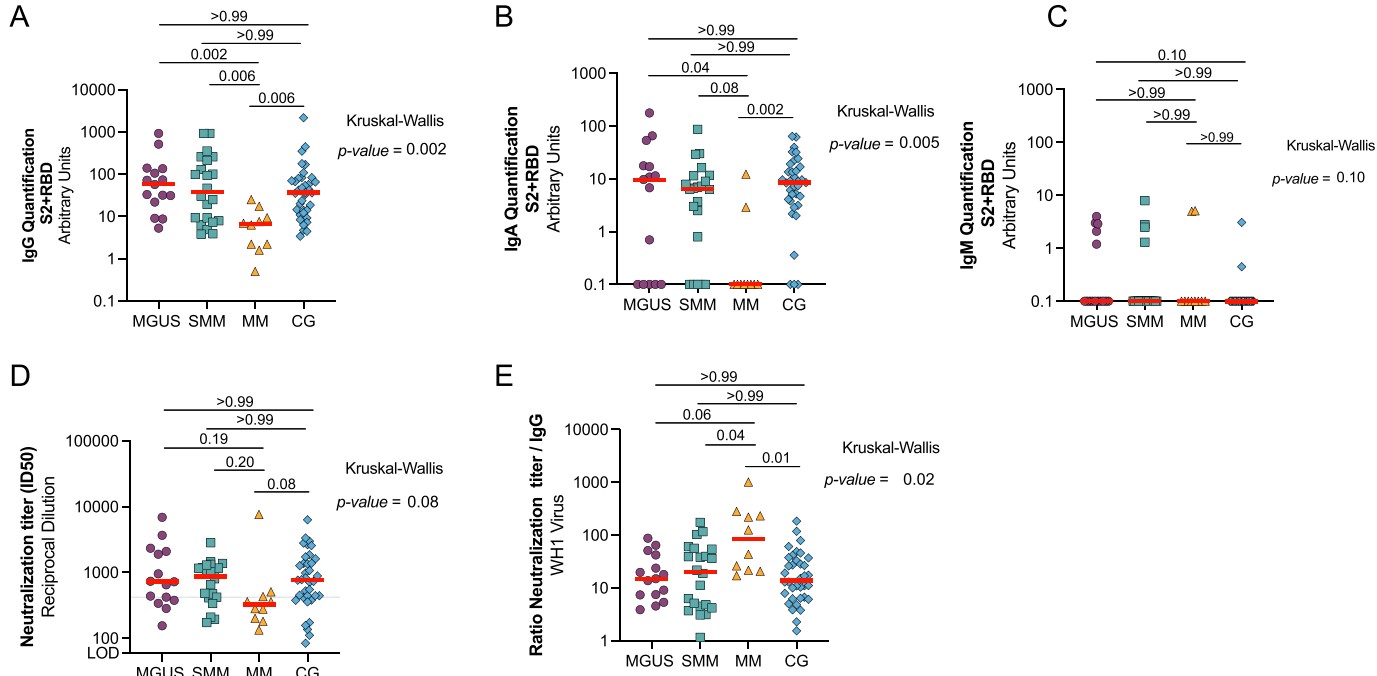

**Figure 2. Comparison of humoral response after 3 mo from mRNA vaccination in uninfected patients with monoclonal gammopathies compared with a control group (CG).**
**(A, B, C, D)** Levels of specific SARS-CoV-2 immunoglobulins IgG (Panel A), IgA (Panel B), and IgM (Panel C) against S2+RBD proteins quantified and neutralizing activity against WH1 (Panel D) from MGUS, SMM, and MM patients and a control group (CG) after 3 mo of vaccine administration. **(D)** Dotted line indicates the 25th percentile of neutralization titer from CG (Panel D). Panel **(E)** Ratio of plasma neutralization titer per total SARS-CoV-2 IgG antibodies. Median values are indicated; *P*-values were obtained from Kruskal–Wallis test for comparison between groups and the post hoc Dunn's multiple comparison's test. Only significant *P*-values are shown. Each symbol represents a participant, and are color-coding according to their disease group (MGUS, purple square; SMM, upper triangle turquoise; MM, lower triangle orange, and CG, blue diamond).
Source data are available for this figure.

Terpos et al, 2021a). Even though MM patients showed low levels of specific SARS-CoV-2 antibodies, these antibodies seemed to be functional because a high ratio of neutralization/total anti–SARS-CoV-2 was observed. Despite we have included only MM patients during the first line of therapy, who may have better preserved immune function compared with more advanced stages of the disease (Bitoun et al, 2021), the levels of neutralizing antibodies remained low, demonstrating that this population would benefit from a booster SARS-CoV-2 vaccine dose (Re et al, 2021 *Preprint*). Importantly and similarly to other studies (Gavriatopoulou et al, 2021; Van Oekelen et al, 2021), we detected significantly higher plasma neutralization capacity in MM individuals who recovered from COVID-19 compared with their uninfected counterparts, highlighting that hybrid immunity elicit stronger immune responses, similar to healthy individuals, even in this immuno-compromised population (Andreano et al, 2021; Crotty, 2021). Indeed, unvaccinated recovered COVID-19 MM patients on active treatment show already a superior antibody response compared with MM subjects after 1 mo complete vaccine schedule (two doses) (Gavriatopoulou et al, 2021).

The humoral response in MGUS and SMM patients has been less studied until now. We observed similar levels of SARS-CoV-2 antibodies and plasma neutralization capacity in MGUS and SMM patients compared to the healthy control group. Similarly, Terpos et al (2021a) did not identify significant differences between MGUS

and controls regarding the development of neutralizing SARS-CoV-2 antibodies after 50 d from SARS-CoV-2 vaccination (Terpos et al, 2021a). However, in this same study, SMM patients achieved more variable vaccine-induced responses and only 61% achieved a clinically relevant antibody response. In our study, MGUS and SMM showed comparable percentages of neutralization activity compared with the control group (around 73–82% in all cases), being this proportion for SMM patients higher than previously described. The lower number of patients included in our study (N = 22 versus N = 38) or the functional assay used may explain the differences observed between both studies.

The underlying causes for suboptimal humoral response to SARS-CoV-2 vaccine in uninfected MM patients may be multifactorial, including disease-related immune dysregulation and the immunosuppression caused by the anti-myeloma therapies. First, myeloma cells can suppress the expansion of normal B-cells and the production of immunoglobulins after SARS-CoV-2 vaccine administration. However, only MM patients showed a suboptimal humoral immune response after vaccination, suggesting that the symptomatic disease plays a crucial role in immunosuppression, whereas the asymptomatic disease may preserve humoral responses. Second, some anti-myeloma therapies could deplete B cells and impair T-cell function, which may hamper also the response to vaccines with poorer neutralizing antibody and cellular immunity responses (Ludwig et al, 2021). Various groups have

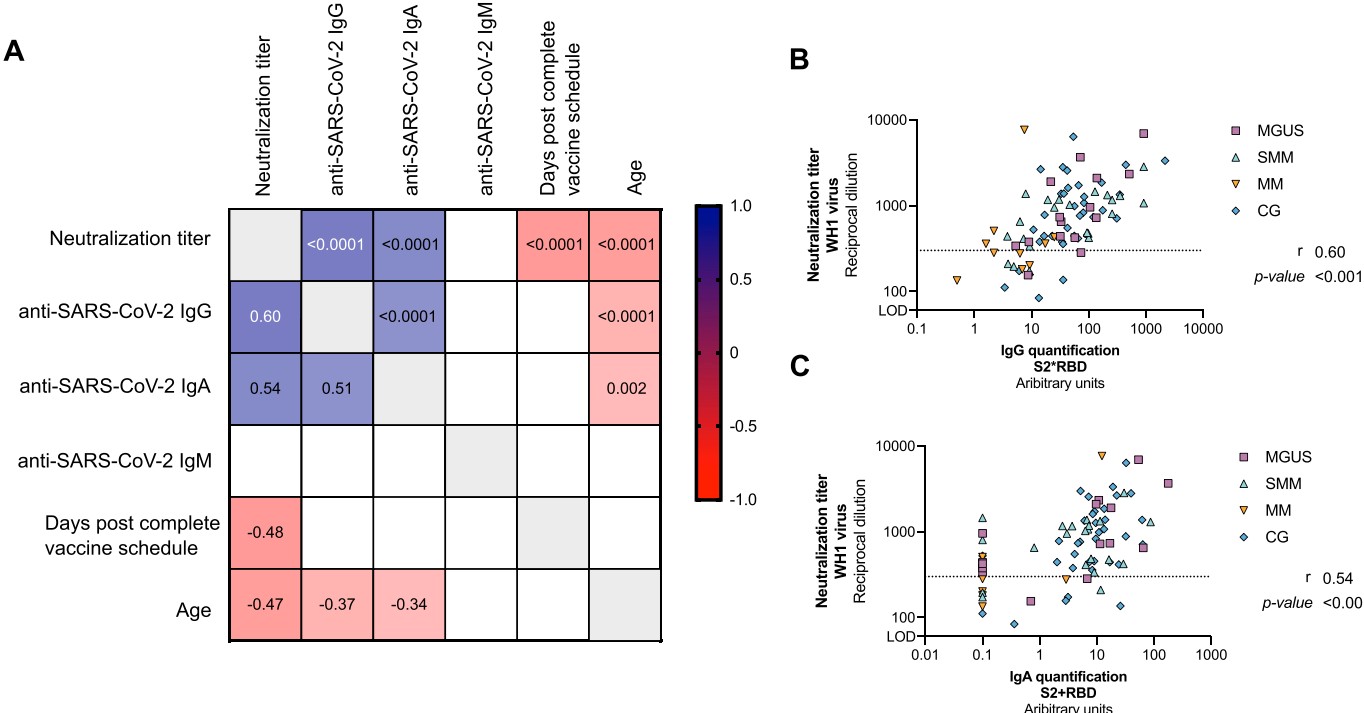

**Figure 3. Correlations between variables.**
Panel **(A)** Correlation matrix of relevant continuous variables including MGUS, SMM, MM, and CG participants. Spearman coefficients are indicated in the lower part of the panel, whereas *P*-values in the upper part. Only significant correlations are plotted (*P* < 0.05). Positive correlations are shown in blue, whereas negative in red. **(B, C)** Detail of the correlation between the levels of SARS-CoV-2–specific IgG (Panel B) and IgA (Panel C) antibodies and neutralization capacity. Each symbol represents a participant, and are color-coding according to their disease group (MGUS, purple square; SMM, upper triangle turquoise; MM, lower triangle orange, and CG, blue diamond). Correlation coefficient and *P*-values were obtained from Spearman correlation.

observed an defective humoral response in patients on daratumumab-based therapies compared with other treatments (Chung et al, 2021; Pimpinelli et al, 2021; Van Oekelen et al, 2021; Terpos et al, 2021a), suggesting a dysfunction of the immune system and relation between plasma cells and bone marrow microenvironment. However, our study and others (Bitoun et al, 2021; Gavriatopoulou et al, 2021), with a reduced number of patients analysed in all cases, did not find this association on patients with daratumumab. Similarly to others (Bitoun et al, 2021), we did not find lower anti–SARS-CoV-2–specific antibodies or neutralization capacity on MM patients with previous HSCT compared with their counterparts. Additional studies with higher number of patients are required to validate these results.

Age may also contribute to a lower immune response to SARS-CoV-2 vaccines among non-cancer uninfected controls older than 65 yr (Collier et al, 2021; Trigueros et al, 2022). In this study, we observed that older age was negatively associated with the levels of anti–SARS-CoV-2 IgG antibodies as well as the neutralization capacity, as other groups have already highlighted (Chung et al, 2021; Terpos et al, 2021b). Because of the older age of most MM patients, this parameter should be taken into account for future analysis.

Immunoparesis has been also associated with an inferior antibody response (Terpos et al, 2021b), and we did confirm these results in our univariate analysis, but was not confirmed in the multivariate validation. In that sense, we observed greater immunoparesis in SMM (77%) and MM (90%) patients compared

with the 35% of MGUS subjects. Despite this high frequency in immunoparesis in SMM group, no significant differences in the levels of anti–SARS-CoV-2 antibodies or plasma neutralization capacity were detected between MGUS, SMM and the healthy control groups. Overall, our results suggest that other factors rather than immunoparesis may contribute to a lower response to vaccination in MM patients. In our cohort, SMM and MM presented similar defects in immune effector cells and B-cell disorders, and the only differential finding between both groups was antimyeloma therapy and lymphopenia.

Interestingly, we observed that mRNA-1273 COVID-19 vaccine was associated to a higher levels of neutralization activity in our multivariate analysis, as previously described (Chung et al, 2021). This result should be further investigated in larger longitudinal studies of patients with MM.

Even though an assessment of the immune response several months after vaccination has been performed, our study has some limitations. First, the relatively small number of participants, especially for the uninfected MM group, limited the multivariate analysis especially for the impact of treatment. In addition, we did not assess specific SARS-CoV-2 cellular responses after vaccination, which could also be used as a correlate of protection (San Segundo et al, 2021).

In conclusion, our study demonstrates that patients suffering from MGUS and SMM did not show significant differences in the plasma neutralization capacity compared with healthy controls,

**Table 2. Univariate and Multivariate analysis.**

| | Univariate | | | Multivariate | | |
|---|---|---|---|---|---|---|
| | Estimated | St error | P-value | Estimated | St error | P-value |
| **Univariate and multivariate analysis for SARS-CoV-2-specific IgG** | | | | | | |
| Group [Reference CG] | | | | | | |
| MGUS | 0.041 | 0.193 | 0.832 | 0.152 | 0.197 | 0.444 |
| SMM | -0.056 | 0.170 | 0.744 | 0.219 | 0.230 | 0.343 |
| MM | -1.016 | 0.225 | <0.0001 | -0.686 | 0.287 | 0.019 |
| Sex (Woman) | -0.119 | 0.156 | 0.446 | | | |
| Age | -0.016 | 0.005 | 0.002 | -0.013 | 0.005 | 0.007 |
| Gammopathy subtype | | | | | | |
| IgG (Kappa or Lambda) | -0.337 | 0.176 | 0.059 | | | |
| IgA (Kappa or Lambda) | 0.129 | 0.250 | 0.607 | | | |
| IgM (Kappa or Lambda) | -0.281 | 0.308 | 0.364 | | | |
| Bence Jones Kappa | -0.242 | 0.508 | 0.635 | | | |
| Bence Jones Lambda | -0.446 | 0.708 | 0.530 | | | |
| Immunoparesis | -0.438 | 0.153 | 0.005 | -0.245 | 0.213 | 0.253 |
| Days post-complete schedule | -0.003 | 0.002 | 0.117 | | | |
| Vaccine kind (Moderna) | 0.157 | 0.161 | 0.335 | | | |
| **Univariate and multivariate analysis for SARS-CoV-2-specific IgA** | | | | | | |
| Group [Reference CG] | | | | | | |
| MGUS | -0.089 | 0.176 | 0.617 | 0.038 | 0.181 | 0.834 |
| SMM | -0.183 | 0.155 | 0.242 | 0.125 | 0.211 | 0.557 |
| MM | -0.735 | 0.205 | <0.001 | -0.370 | 0.263 | 0.164 |
| Sex (Woman) | -0.080 | 0.135 | 0.555 | | | |
| Age | -0.012 | 0.004 | 0.004 | -0.009 | 0.004 | 0.030 |
| Gammopathy subtype | | | | | | |
| IgG (Kappa or Lambda) | -0.316 | 0.151 | 0.040 | | | |
| IgA (Kappa or Lambda) | -0.087 | 0.214 | 0.686 | | | |
| IgM (Kappa or Lambda) | -0.374 | 0.264 | 0.161 | | | |
| Bence Jones Kappa | 0.079 | 0.436 | 0.856 | | | |
| Bence Jones Lambda | -0.890 | 0.608 | 0.147 | | | |
| Immunoparesis (yes) | -0.436 | 0.130 | 0.001 | -0.317 | 0.195 | 0.109 |
| Days post-complete schedule | -0.002 | 0.002 | 0.280 | | | |
| Vaccine kind (Moderna) | 0.202 | 0.139 | 0.151 | | | |

**Table 2.  Continued**

Univariate and multivariate analysis for neutralization

| | Univariate | | | Multivariate | | |
|---|---|---|---|---|---|---|
| | Estimated | St error | P-value | Estimated | St error | P-value |
| Group [Reference CG] | | | | | | |
| MGUS | 0.010 | 0.129 | 0.941 | 0.003 | 0.108 | 0.979 |
| SMM | −0.040 | 0.114 | 0.725 | −0.105 | 0.138 | 0.449 |
| MM | −0.415 | 0.150 | 0.007 | −0.507 | 0.173 | 0.005 |
| Sex (Woman) | −0.078 | 0.099 | 0.433 | | | |
| Age | −0.013 | 0.003 | <0.0001 | −0.009 | 0.003 | 0.001 |
| Gammopathy subtype | | | | | | |
| IgG (Kappa or Lambda) | −0.104 | 0.113 | 0.362 | | | |
| IgA (Kappa or Lambda) | 0.055 | 0.161 | 0.734 | | | |
| IgM (Kappa or Lambda) | −0.187 | 0.198 | 0.346 | | | |
| Bence Jones Kappa | −0.104 | 0.326 | 0.749 | | | |
| Bence Jones Lambda | −0.331 | 0.455 | 0.453 | | | |
| Immunoparesis (yes) | −0.203 | 0.097 | 0.040 | 0.043 | 0.118 | 0.721 |
| Days post-complete schedule | −0.004 | 0.001 | <0.0001 | −0.005 | −0.005 | <0.0001 |
| Vaccine kind (Moderna) | 0.167 | 0.102 | 0.107 | 0.258 | 0.093 | 0.007 |

Significant P-values are indicated in bold and all P-values are in italic.

and SARS-CoV-2 booster vaccines should be administrated following the recommendations for the general population. In contrast, MM patients in first line of therapy have a blunted antibody response after 3 mo from complete vaccine administration, and tailored booster campaigns of the vaccine-induced immune responses should be considered in these cancer patients to adapt their SARS-CoV-2 vaccination calendar to their immune needs.

# Materials and Methods

### Study population

A cross-sectional study (VAC-COV-GM-HMAR) was conducted in patients with monoclonal gammopathies at Hospital del Mar to assess the efficacy of vaccination against SARS-CoV-2, after having received the complete vaccination schedule dose according to BNT162b2 (Pfizer-BioNTech) and mRNA-1273 COVID-19 (Moderna) schedules (median of 3.9 IQR [2.2–5] months post-vaccine). 59 patients suffering monoclonal gammopathies consecutively visited during 2 mo at the outpatient haematological consult were eligible to entry into the study. Treated patients received only one line of therapy before vaccination.

Results were compared with a control group, which included 36 uninfected and 58 infected individuals without haematological malignancies, belonging to the *King cohort extension* (*N* = 47) and CoronAVI@S (N = 47) studies. Post-vaccine samples were selected among individuals vaccinated also with BNT16b2 and mRNA-1273 COVID-19.

The VAC-COV-GM-HMAR, *King cohort extension*, and CoronAVI@S studies were approved by the Ethics Committee Boards from the Hospital del Mar (HMAR), the Hospital Universitari Germans Trias i Pujol (HUGTIP), and the Institut Universitari d'Investigació en Atenció Primària (IDIAP), respectively (HMAR/2021/9913/I, HUGTiP/PI-20-217, and IDIAP/20-116P) and were conducted in accordance with the Declaration of Helsinki. All patients provided written informed consent before starting the study.

### Determination of anti–SARS-CoV-2 antibodies

The presence of anti–SARS-CoV-2 antibodies against Spike S2 Subunit+ Spike protein receptor binding domain (S2+RBD) or nucleocapsid protein (NP) in plasma samples was evaluated using an in-house developed sandwich-ELISA, as previously described (Massanella et al, 2021). Briefly, Nunc MaxiSorp ELISA plates (Cat. no. M9410-1CS; Sigma-Aldrich) were coated overnight at 4°C with 50 ng/ml of capture antibody (anti-6xHis antibody, clone HIS.H8; Cat. no. MA1-21315; Thermo Fisher Scientific) at 2 $\mu$g/ml in PBS. After washing, plates were blocked for 2 h at room temperature using PBS/1% of BSA (Cat. no. 130-091-376; Miltenyi Biotech). Then, 50 $\mu$l of the following SARS-CoV-2–derived antigens diluted in blocking buffer were added: Spike (S2) (0.9 $\mu$g/ml, Cat. no. 40590-V08B), receptor binding domain (RBD, Cat. no. 40592-V08B) (0.3 $\mu$g/ml), or nucleocapsid protein (NP, 40588-V08B) (1 $\mu$g/ml) (Sino Biologicals) and incubated overnight at 4°C. Each plasma sample was evaluated in duplicated at dilution ranging from 1/100 to 1/50,000 in blocking buffer for each antigen. Diluted samples were incubated at room temperature for 1 h. Antigen-free wells were also assayed in parallel for each sample in the same plate to evaluate sample background. Serial dilutions of a positive plasma sample were used as standard. A pool of 10 SARS-CoV-2–negative plasma samples, collected before June 2019, were included as negative control. The following reagents were used as secondary antibodies: HRP-conjugated (Fab)2 Goat anti-human IgG (Fc specific, Cat. no. 109-036-098) (1/20,000), Goat anti-human IgM (1/10,000, Cat. no. 109-036-129), and Goat anti-human IgA ($\alpha$ chain specific, Cat. no. 109-036-011) (1/10,000) (all from Jackson Immunoresearch). Secondary antibodies were incubated for 30 min at room temperature. After washing, plates were revealed using o-Phenylenediamine dihydrochloride (OPD, Sigma-Aldrich, Cat. no. P8787) and the enzymatic reaction was stopped with 4N of $H_2SO_4$ (Sigma-Aldrich). The signal was analysed as the optical density (OD) at 492 nm with noise correction at 620 nm. The specific signal for each antigen was calculated after subtracting the background signal obtained for each sample in antigen-free wells. Values are plotted into the standard curve. Standard curve was calculated by plotting and fitting the log of standard dilution (in arbitrary units) versus response to a four-parameter equation in Prism 8.4.3 (GraphPad Software).

### Pseudovirus neutralization assay

Neutralization assay was performed using SARS-CoV-2.SctΔ19 WH1 pseudovirus as previously described (Trinité et al, 2021; Pradenas et al, 2022a). Briefly, SARS-CoV-2.SctΔ19 WH1 and B.1.617.2/Delta were generated (Geneart) from the full protein sequence of the original SARS-Cov-2 isolate Wuhan-Hu-1 (WH1) spike sequence, with the deletion of the last 19 amino acids in C-terminal (Ou et al, 2020), human-codon optimized and inserted into pcDNA3.1(+). HIV reporter pseudoviruses expressing SARS-CoV-2 S protein and Luciferase were generated using the defective HIV plasmid pNL4-3.Luc.R-.E—obtained from the NIH AIDS Reagent Program (Connor et al, 1995). Expi293F cells were transfected using ExpiFectamine293 Reagent (Thermo Fisher Scientific) with pNL4-3.Luc.R-.E– and SARS-CoV-2.SctΔ19 (WH1, B.1.617.2/Delta), at an 8:1 ratio, respectively. Control pseudoviruses were obtained by replacing the S protein expression plasmid with a VSV-G protein expression plasmid as reported (Sánchez-Palomino et al, 2011). Supernatants were harvested 48 h after transfection, filtered at 0.45 $\mu$m, frozen and titrated on HEK293T cells overexpressing WT human ACE-2 (Integral Molecular). Neutralization assays were performed in duplicate. Briefly, in Nunc 96-well cell culture plates (Cat. no. 165305; Thermo Fisher Scientific), 200 TCID50 of pseudovirus were preincubated with threefold serial dilutions (1/60–1/14,580) of heat-inactivated plasma samples for 1 h at 37°C. Then, 2 × 10⁴ HEK293T/hACE-2 cells treated with DEAE-Dextran (Cat. no. D9885; Sigma-Aldrich) were added. Results were read after 48 h using the EnSight Multimode Plate Reader and BriteLite Plus Luciferase reagent (Cat. no. 6066761; Perkin Elmer). The values were normalized, and the ID50 (the reciprocal dilution inhibiting 50% of the infection) was calculated by plotting and fitting the log of plasma dilution versus response to a four-parameter equation in Prism 8.4.3 (GraphPad Software). This neutralization assay had been previously validated in a large subset of samples (Trinité et al, 2021; Pradenas et al,

2022a). The lower limit of detection was 60 and the upper limit was 14,580 (reciprocal dilution).

## Statistical analysis

Continuous variables were described using medians and the interquartile range (IQR), whereas categorical factors were reported as percentages. Quantitative variables were compared using the Mann–Whitney test and proportions using the chi-squared test for comparison between two groups, and nonparametric Kruskal–Wallis test for comparison between all groups. Correlations between continuous variables were assessed with Spearman's rank correlation coefficient. The impact of each variable to the levels of specific SARS-CoV-2 IgG or IgA antibodies and neutralization capacity was assessed via linear regression models including all patients with monoclonal gammopathies and the control groups, using the latter as reference. A multivariate model was constructed based on significant variables on the univariate analysis with an inclusion criterion of $P < 0.1$. Statistical analyses were performed with Prism 9.1.2 (GraphPad Software) and R (4.1.2). Statistical significance was determined when $P \leq 0.05$.

# Data Availability

The authors confirm that the data supporting the findings of this study are available within the article and its supplementary materials. The data are available on request from the corresponding author (E Abella/M Massanella).

# Supplementary Information

# Acknowledgements

We are deeply grateful to all participants who participated in this study. We also thank the technical staff of Hospital del Mar, Direcció d'Atenció Primària de la Metropolitana Nord and AIDS Figth foundation for sample collection, and staff of IrsiCaixa for sample processing (L Ruiz, R Ayen, L Gomez, C Ramirez, M Martinez, T Puig). We thank CERCA Programme/Generalitat de Catalunya for institutional support and the Foundation Dormeur. This work was partially funded by Grifols, the Departament de Salut of the Generalitat de Catalunya (grant DSL0016 to J Blanco and Grant DSL015 to J Carrillo), the Spanish Health Institute Carlos III (Grant PI17/01518 and PI20/00093 to J Blanco and PI18/01332 to J Carrillo), Fundació Gloria Soler, and the crowdfunding initiatives https://www.yomecorono.com, BonPreu/Esclat and Correos. M Trigueros was supported by a doctoral fellowship from the Departament de Salut from Generalitat de Catalunya (SLT017/20/000095). F Muñoz-Lopez is supported by a doctoral grant from Sorigué Foundation. E Pradenas was supported by a doctoral grant from ANID, Chile: Grant 72180406. M Massanella was granted with RYC2020-028934-I/AEI/10.13039/501100011033 from Spanish Ministry of Science and Innovation and State Research Agency, MCIN/AEI/10.13039/501100011033 and the European Social Fund "investing in your future."

## Author Contributions

E Abella: conceptualization, supervision, funding acquisition, methodology, writing—original draft, and project administration.
M Trigueros: data curation, methodology, and writing—review and editing.
E Pradenas: data curation, methodology, and writing—review and editing.
F Muñoz-Lopez: formal analysis.
F Garcia-Pallarols: resources, methodology, and writing—review and editing.
R Ben Azaiz Ben Lahsen: resources, data curation, methodology, project administration, and writing—review and editing.
B Trinité: data curation, formal analysis, methodology, and writing—review and editing.
V Urrea: data curation, formal analysis, supervision, and writing—review and editing.
S Marfil: data curation, formal analysis, and writing—review and editing.
C Rovirosa: data curation, formal analysis, and writing—review and editing.
T Puig: data curation, formal analysis, and writing—review and editing.
E Grau: data curation, formal analysis, supervision, project administration, and writing—review and editing.
A Chamorro: resources, data curation, and writing—review and editing.
R Toledo: data curation, methodology, and writing—review and editing.
M Font: resources, data curation, and writing—review and editing.
D Palacin: resources, data curation, and writing—review and editing.
F Lopez-Segui: investigation and writing—review and editing.
J Carrillo: resources, data curation, formal analysis, funding acquisition, and writing—review and editing.
N Prat: resources, data curation, project administration, and writing—review and editing.
L Mateu: resources, data curation, formal analysis, methodology, and writing—review and editing.
B Clotet: conceptualization, data curation, funding acquisition, methodology, and writing—review and editing.
J Blanco: resources, data curation, formal analysis, funding acquisition, and writing—review and editing.
M Massanella: conceptualization, data curation, formal analysis, supervision, funding acquisition, investigation, methodology, project administration, and writing—original draft, review, and editing.

## Conflict of Interest Statement

J Blanco and J Carrillo reports personal fees from Albajuna Therapeutics, outside the submitted work.

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
