## [Reviewer comments · Life Science Alliance]

Life Science Alliance

Efficacy of SARS-CoV-2 vaccination in patients with monoclonal gammopathies - cross sectional study

Eugenia Abella, Macedonia Trigueros, Edwards Pradenas, Francesc Garcia-Pallarols, Randa Ben Azaiz Ben Lahsen, Benjamin Trinité, Victor Urrea, Silvia Marfil, Carla Roviroso, Teresa Puig, Eulàlia Grau, Anna Chamorro, Ruth Toledo, Marta Font, Dolors Palacín, Francesc Lopez-Segui, Jorge Carrillo, Nuria Prat, Lourdes Mateu, Bonaventura Clotet, Julia Blanco, and Marta Massanella

DOI: <https://doi.org/10.26508/lsa.202201479>

Corresponding author(s): Marta Massanella, IrsiCaixa AIDS Research Institute and Eugenia Abella, Department of Hematology, Hospital del Mar-IMIM

Review Timeline:

Submission Date:	2022-04-11
Editorial Decision:	2022-05-27
Revision Received:	2022-07-04
Editorial Decision:	2022-07-19
Revision Received:	2022-07-20
Accepted:	2022-07-20

Scientific Editor: Novella Guidi

Transaction Report:

May 27, 2022

Re: Life Science Alliance manuscript #LSA-2022-01479

Dr. Marta Massanella
IrsiCaixa AIDS Research Institute
Hospital Germans Trias i Pujol
Carretera del Canyet s/n
Badalona 08916
Spain

Dear Dr. Massanella,

Thank you for submitting your manuscript entitled "Efficacy of SARS-CoV-2 vaccination in patients with monoclonal gammopathies - cross sectional study" to Life Science Alliance. The manuscript was assessed by expert reviewers, whose comments are appended to this letter. We invite you to submit a revised manuscript addressing the Reviewer comments.

Thank you for this interesting contribution to Life Science Alliance. We are looking forward to receiving your revised manuscript.

Sincerely,

B. MANUSCRIPT ORGANIZATION AND FORMATTING:

Reviewer #1 (Comments to the Authors (Required)):

In this article the authors quantified immunoglobulins IgG, IgA and IgM in healthy patients versus patients with gammopathies after vaccination. they could observe that patients with MGUS or SMM produce similar level than healthy control in response to mRNA vaccine. However MM patients produce less IgG and IgA in response to vaccine. However the neutralization titer was not significantly reduced in MM patients.

In the first figure authors displayed a group of patients MGUS with anti-NP SARS-CoV-2 IgG. this group contain only two patients which is too low in order to perform meaningful statistical test. the authors should remove this group or implement at least a third patient.

the authors forgot to include healthy control with or without anti-NP SARS-CoV-2 IgG. Since the authors used a log scale it is very striking that SARS-CoV-2 IgA or IgM quantification in MGUS patient group seems to have two clear subgroups of patients. Did the author check if it was sex related or if any other parameter could explain that 5 MGUS patients produce much more IgM and does these patients produce more IgA? I do think the authors could have explore more these 5 patients and what are their characteristic.

I do think the authors cannot conclude anything about infected MGUS patients since the cohort contain only 2 patients. The neutralization panel do not give so much information since there is no healthy control. Except that SARS-CoV-2 infection in MM patients induce better neutralization than the vaccine

Figure 2 the authors should display that MGUS and SMM patients IgG, IgA do not differ from healthy patients since their goal is to claim that these patients respond properly to vaccination in term of antibody production.

In this figure also the authors show that very few patients produce IgM after vaccination but are these patients producing more SARS-CoV-2 IgA or IgG? Sadly, the neutralization clearly shows no significant differences between healthy control, MGUS, SMM and MM in term of neutralization titer. The authors used a ratio of neutralization titer/ IgG in order to observe a difference. Figure 3, this figure just confirms that neutralization correlate with SARS-CoV-2 IgG and IgA quantity and decrease with time post vaccination and age.

I do think the authors need to be more rigorous with the figure 1 by including controls and more than 2 patients for MGUS infected patients if they want to state anything concerning this type of patients. the authors overstate a bit their results when they say.

Reviewer #2 (Comments to the Authors (Required)):

The manuscript by Abella et al. is a retrospective study of the neutralizing antibody titers (spike protein) to Covid vaccination using mRNA based (Pfizer and Moderna) vaccines. The authors made some interesting observations most of which are confirmatory. The manuscript is well-written although I think it could be improved.

Methodology: the method in which the patients were tested needs to be expanded. It would be helpful to note at what time point were the patients tested for the antibody titers after vaccination. This is especially important since the authors looked at both IgA and IgG titers since IgA is felt to be the earlier response.

The observation that time from vaccination is negatively associated with titer is an important one. In fact, not only is the peak titer response may be less but the duration may be decreased in these patients. It would be helpful to know when the titer is more likely to drop below the protection limits.

Table 1 and 2 can be combined together.

Table 3a and 3b. the title is the level of specific IgA/IgG but not levels or concentration was actually given. The title should be relabelled to reflect the information given.

We would like to thank the reviewers for their revisions, which have improved the quality of the manuscript. We have addressed all suggestions by the reviewers, and these have been incorporated and highlighted in the new version of the manuscript. A new figure (heatmap) has been created for reviewers' consideration to be included in the manuscript.

Reviewer #1 (Comments to the Authors (Required)):

In this article the authors quantified immunoglobulins IgG, IgA and IgM in healthy patients versus patients with gammopathies after vaccination. They could observe that patients with MGUS or SMM produce similar level than healthy control in response to mRNA vaccine. However MM patients produce less IgG and IgA in response to vaccine. However the neutralization titer was not significantly reduced in MM patients.

In the first figure authors displayed a group of patients MGUS with anti-NP SARS-CoV-2 IgG. this group contain only two patients which is too low in order to perform meaningful statistical test. the authors should remove this group or implement at least a third patient.

We agree with the reviewer regarding the limited significance of only 2 participants. We have removed the statistics from Figure 1 and the main text (Page 8), however we have kept the anti-NP+ MGUS group in Figure 1 to be transparent and show the data before to exclude them in the following analysis (Figure 2).

The authors forgot to include healthy control with or without anti-NP SARS-CoV-2 IgG.

We now have included a healthy control group in Figure 1, which was also subclassified in two groups: previously infected and uninfected. Anti-NP+ MM participants showed significantly higher levels of circulating SARS-CoV-2 specific IgG and IgA antibodies against S2+RBD compared to their anti-NP- counterparts ($p=0.003$ and $p=0.04$, respectively, Figure 1A and B). However, anti-NP+ MM group showed significantly lower SARS-COV-2 specific IgG and IgA antibodies compared to the control infected group ($p=0.02$ and $p=0.04$, respectively). Plasma from anti-NP+ MM patients showed statistically increased levels of neutralization compared to their uninfected counterparts (anti-NP-) and uninfected controls ($p=0.01$ and $p=0.04$, respectively Figure 1D), while similar levels of neutralization were observed between anti-NP+ MM and infected control group ($p=0.4$). These new results have been included in the manuscript (Page 8).

Since the authors used a log scale it is very striking that SARS-CoV-2 IgA or IgM quantification in MGUS patient group seems to have two clear subgroups of patients. Did the author check if it was sex related or if any other parameter could explain that 5 MGUS patients produce much more IgM and does these patients produce more IgA? I do think the authors could have explore more these 5 patients and what are their characteristic.

To address the reviewer concerns, we have performed additional analysis. Despite the levels of SARS-CoV-2 specific IgM antibodies were positive in 5 MGUS individuals, they remained very low and almost undetectable. Some individuals presented higher levels of IgA. However, no special characteristic in terms of sex, age or other demographic parameter analyzed were found between the subgroups of patients producing SARS-CoV-2 specific IgM antibodies or producing higher levels of SARS-CoV-2 specific IgA antibodies (data not shown). In addition, no correlation was found between the levels of SARS-CoV-2 specific IgM and IgG or IgA antibodies (data not shown).

We performed a heat-map graph for visualization purposes, which shows the levels of SARS-CoV-2 specific IgG, IgA and IgM at the individual level. In general, detectable levels of IgM were associated to

detectable higher levels of IgG or IgA antibodies (Figure A, see below, for the reviewer's consideration), but no unique pattern was observed.

Figure A: Heatmap of SARS-CoV-2 specific IgG, IgA and IgM in uninfected (left panel) and infected (right panel) participants, including MGUS (purple), SMM (turquoise), MM (yellow) and control group (CG, blue) individuals.

I do think the authors cannot conclude anything about infected MGUS patients since the cohort contain only 2 patients. The neutralization panel do not give so much information since there is no healthy control. Except that SARS-CoV-2 infection in MM patients induce better neutralization than the vaccine.

With the inclusion of previously infected control group, we believe that the comparison with anti-NP+ MM group is interesting. While anti-NP+ MM individuals show statistically significant lower SARS-CoV-2 specific IgG and IgA, plasma neutralization capacity remains similar between groups. These results have been included in the manuscript (Page 8).

Figure 2 the authors should display that MGUS and SMM patients IgG, IgA do not differ from healthy patients since their goal is to claim that these patients respond properly to vaccination in term of antibody production.

In Figure 2, we only included significant differences between groups, and this is the reason why we did not indicate the p-values between MGUS and SMM and CG group. As suggested by the reviewer, we have included the p-values in the new version of Figure 2.

In this figure also the authors show that very few patients produce IgM after vaccination but are these patients producing more SARS-CoV-2 IgA or IgG? Sadly, the neutralization clearly shows no significant differences between healthy control, MGUS, SMM and MM in term of neutralization titer. The authors used a ratio of neutralization titer/ IgG in order to observe a difference.

As noted by the reviewer, we only found a trend in the plasma neutralization capacity between MM and CG uninfected groups ($p=0.08$), while no differences were observed between MGUS, SMM and CG individuals. In contrast, we observed significant differences when we used the ratio of neutralization / SARS-CoV-2 specific IgG antibodies. We seek to evaluate the quality of the response with this ratio. Indeed, we have previously used this ratio in peer-reviewed research articles (Trigueros *et al.*, Age and Ageing 2022 and Pradenas *et al.*, Frontiers in Immunology 2022). We believe this ratio is especially important in vulnerable populations, such as cancer patients or elder individuals (Trigueros *et al.*, Age and Ageing 2022). We have included both references in the main text.

Figure 3, this figure just confirms that neutralization correlate with SARS-CoV-2 IgG and IgA quantity and decrease with time post vaccination and age.

As reviewer noted, Figure 3 is just a confirmatory analysis.

I do think the authors need to be more rigorous with the figure 1 by including controls and more than 2 patients for MGUS infected patients if they want to state anything concerning this type of patients. the authors overstate a bit their results when they say.

As suggested by the reviewer, we include the control group and removed the statistics and conclusion between MGUS anti-NP- and anti-NP+ participants, due to the low number of individuals in the latter group (Page 8).

Reviewer #2 (Comments to the Authors (Required)):

The manuscript by Abella *et al.* is a retrospective study of the neutralizing antibody titers (spike protein) to Covid vaccination using mRNA based (Pfizer and Moderna) vaccines. The authors made some interesting observations most of which are confirmatory. The manuscript is well-written although I think it could be improved.

We thank the reviewer for the positive feedback and the recommendations to improve our manuscript.

Methodology: the method in which the patients were tested needs to be expanded. It would be helpful to note at what time point were the patients tested for the antibody titers after vaccination. This is especially important since the authors looked at both IgA and IgG titers since IgA is felt to be the earlier response.

Methodology details regarding the quantification of SARS-CoV-2 specific antibodies (IgG, IgA and IgM) and the pseudovirus neutralization assay are included in supplemental material.

We performed all assays a median 3.9 IQR [2.2-5] months from complete vaccine schedule. Control group were matched for time post-complete vaccine schedule. This information can be found in Material and Methods section (Page 6), results section (Page 8) and Table 1.

The observation that time from vaccination is negatively associated with titer is an important one. In fact, not only is the peak titer response may be less but the duration may be decreased in these patients. It would be helpful to know when the titer is more likely to drop below the protection limits.

We agree with the reviewer that it would be interesting to evaluate humoral responses at longer time points after complete vaccine schedule. However, the study was not design to obtain a follow up visit. Despite this missing data, MM uninfected group may be already below the protection limits at month 3 post-vaccination (2 doses), and booster doses might necessary to reach a protection threshold. Since there are no differences in humoral responses between MGUS and SMM groups with uninfected CG, we could expect a similar drop of humoral responses in all groups and, MGUS and SMM patients could follow similar recommendations for vaccine booster doses as the general population.

Table 1 and 2 can be combined together.

We have combined table 1 and 2 as reviewer suggested.

Table 3a and 3b. the title is the level of specific IgA/IgG but not levels or concentration was actually given. The title should be relabelled to reflect the information given.

We thank the reviewer to notice this mistake. We have modified title of the Table.

July 19, 2022

RE: Life Science Alliance Manuscript #LSA-2022-01479R

Dr. Marta Massanella
IrsiCaixa AIDS Research Institute
Hospital Germans Trias i Pujol
Carretera del Canyet s/n
Badalona 08916
Spain

Dear Dr. Massanella,

Thank you for submitting your revised manuscript entitled "Efficacy of SARS-CoV-2 vaccination in patients with monoclonal gammopathies - cross sectional study". We would be happy to publish your paper in Life Science Alliance pending final revisions necessary to meet our formatting guidelines.

- the 1st supplemental material file is supplemental data; please incorporate it in the Materials and Methods section in the main manuscript text
- please add ORCID ID for secondary corresponding author; you should have received instructions on how to do so
- please consult our manuscript preparation guidelines <https://www.life-science-alliance.org/manuscript-prep> and make sure your manuscript sections are in the correct order
- please add an abstract in our system
- please add the Twitter handle of your host institute/organization as well as your own or/and one of the authors in our system
- please add a conflict of interest statement to the main manuscript text
- please add your supplementary figure legends and table legends to the figure legend section of your main manuscript text

A. FINAL FILES:

B. MANUSCRIPT ORGANIZATION AND FORMATTING:

Sincerely,

Reviewer #1 (Comments to the Authors (Required)):

The authors replied to all my comments and improved their manuscript. I do recommend to accept the revised version of the article for publication.

July 20, 2022

RE: Life Science Alliance Manuscript #LSA-2022-01479RR

Dr. Marta Massanella
IrsiCaixa AIDS Research Institute
Hospital Germans Trias i Pujol
Carretera del Canyet s/n
Badalona 08916
Spain

Dear Dr. Massanella,

Thank you for submitting your Research Article entitled "Efficacy of SARS-CoV-2 vaccination in patients with monoclonal gammopathies - cross sectional study". It is a pleasure to let you know that your manuscript is now accepted for publication in Life Science Alliance. Congratulations on this interesting work.

DISTRIBUTION OF MATERIALS:

Again, congratulations on a very nice paper. I hope you found the review process to be constructive and are pleased with how the manuscript was handled editorially. We look forward to future exciting submissions from your lab.

Sincerely,
